# Data-driven head motion correction for PET using time-of-flight and positron emission particle tracking techniques

Tasmia Rahman Tumpa[1,2]☯, Shelley N. Acuff[1], Jens Gregor[2]☯, Yong Bradley[1], Yitong Fu[1], Dustin R. Osborne🄳[1]☯*

1 Molecular Imaging & Translational Research, University of Tennessee Graduate School of Medicine, Knoxville, TN, United States of America, 2 The University of Tennessee: Electrical Engineering and Computer Science, Knoxville, TN, United States of America

☯ These authors contributed equally to this work.
* dosborne@utmck.edu

**Data Availability Statement:** For this study, all data as it pertains to the manuscript is contained within the manuscript itself. For access to patient images, data cannot be openly shared due to IRB

## Abstract

### Objectives

Positron emission tomography (PET) is susceptible to patient movement during a scan. Head motion is a continuing problem for brain PET imaging and diagnostic assessments. Physical head restraints and external motion tracking systems are most commonly used to address to this issue. Data-driven methods offer substantial advantages, such as retroactive processing but typically require manual interaction for robustness. In this work, we introduce a time-of-flight (TOF) weighted positron emission particle tracking (PEPT) algorithm that facilitates fully automated, data-driven head motion detection and subsequent automated correction of the raw listmode data.

### Materials methods

We used our previously published TOF-PEPT algorithm Dustin Osborne et al. (2017), Tasmia Rahman Tumpa et al., Tasmia Rahman Tumpa et al. (2021) to automatically identify frames where the patient was near-motionless. The first such static frame was used as a reference to which subsequent static frames were registered. The underlying rigid transformations were estimated using weak radioactive point sources placed on radiolucent glasses worn by the patient. Correction of raw event data were achieved by tracking the point sources in the listmode data which was then repositioned to allow reconstruction of a single image. To create a "gold standard" for comparison purposes, frame-by-frame image registration based correction was implemented. The original listmode data was used to reconstruct an image for each static frame detected by our algorithm and then applying manual landmark registration and external software to merge these into a single image.

### Results

We report on five patient studies. The TOF-PEPT algorithm was configured to detect motion using a 500 ms window. Our event-based correction produced images that were visually

restrictions, however, data may be accessed by contacting the University of Tennessee Graduate School of Medicine Institutional Review Board for researchers that meet the criteria for access to HIPAA protected and confidential data. The UTGSM IRB does not have a non-individual email address, however, their physical contact information is provided below: The Institutional Review Board (IRB) UTHSC Graduate School of Medicine 3rd Floor, U76 1924 Alcoa Highway Knoxville, TN 37920 Phone: 865-305-6892.

**Funding:** The author(s) received no specific funding for this work.

**Competing interests:** The authors have declared no competing interests exist.

free of motion artifacts. Comparison of our algorithm to a frame-based image registration approach produced results that were nearly indistinguishable. Quantitatively, Jaccard similarity indices were found to be in the range of 85-98% for the former and 84-98% for the latter when comparing the static frame images with the reference frame counterparts.

## Discussion

We have presented a fully automated data-driven method for motion detection and correction of raw listmode data. Easy to implement, the approach achieved high temporal resolution and reliable performance for head motion correction. Our methodology provides a mechanism by which patient motion incurred during imaging can be assessed and corrected post hoc.

## 1 Introduction

Positron emission tomography (PET) is a diagnostic nuclear medicine imaging procedure that allows metabolic activity to be studied. Quantitative and qualitative PET assessment is affected by patient movement resulting from respiratory, cardiac, head, and full-body motion. Artifacts from motion include blurred images as well as inaccurate standardized uptake values that propagate into standard clinical analysis and time activity curves used in more complex kinetic modeling analysis. Head motion correction is of critical concern in brain PET imaging where even restricted movement can negatively affect both visual image quality and quantitative diagnostic assessments of the small structures of the brain. A recent paper by Kyme et al. [1] outlined problems with motion and reviewed motion estimation and correction methods for different modalities. Physical head restraints, such as thermoplastic masks [2], head holders, and vacuum-lock bags [3], can be used to minimize the effects of motion artifacts but are not capable of fully eliminating them, and some are uncomfortable or intolerable for patients. For dementia patients, they can also have a negative psychological impact and should be avoided [4]. The World Health Organization predicts that the number of dementia patients will increase to 139 million from 55 million by 2050 [5]. With the rising number of dementia cases and growing need for beta amyloid imaging, other methods are needed to ensure high quality diagnostic images.

Several external motion tracking systems have been introduced to date based on a variety of technologies including position sensitive detectors [6], optical imaging [7–9], stereo vision [10] and the Microsoft Kinect [11]. The motion information is used to perform either frame based image registration [12] or event based correction [9, 12–25]. Frame based image registration divides the listmode data into a sequence of motion-free frames based on the tracked motion. Images are reconstructed for each frame of data, aligned with a reference image frame, and then summed together to create the final image volume. Event based correction, on the other hand, reposition the individual lines of response (LOR) thereby allowing a single image to be reconstructed from all the raw data. Although external motion tracking systems are effective and can process several frames per seconds, they do not allow retrospective corrections, and widespread use is limited due to the costs associated with initial setup, training, regular maintenance, etc.

Fully data-driven methods do not depend on external hardware or electronics and can correct data from any imaging system post hoc. Typically, these methods have used frame-based

image registration where data is sorted into a sequence of arbitrary, short duration frames. Image registration has been achieved by optimizing a similarity criterion, e.g., mutual information [16–18], cross-correlation [16, 17, 19], sum of absolute differences [19, 20], or standard deviation of the ratio of two image volumes [19, 20]. There is a trade-off between the performance and choice of frame duration. To reduce image noise and improve registration accuracy, longer duration frames are favorable but may lead to intra-frame motion going undetected. Conversely, shorter duration frames are computationally more expensive to handle due to more images having to be reconstructed and processed, and the images will suffer from increased noise. Prior motion information can help determine frame durations.

Recent papers have proposed data-driven approaches that detect motion directly from list-mode or sinogram data and thus prior to image reconstruction. Feng et al. [21] proposed a center-of-mass (COM) method for extracting motion information from rebinned sinogram data. Thielemans et al. [22] applied principal component analysis to dynamic sinogram frames. Schleyer et al. [23] used both principal component and spatial displacement analysis. Motion correction for these methods was in all cases still limited to frame-based reconstructed image registration. Lu et al. [24, 25] recently presented a multi-step COM-based approach that used frame-based reconstructed image registration to obtain a transformation matrix that was then applied to perform event-based correction. In previous work on respiratory motion correction [26–28], we introduced a time-of-flight (TOF) weighted positron emission particle tracking (PEPT) algorithm for motion tracking which we compared against the state-of-the-art from the literature [24, 25]. In this paper, we present a novel, fully automated methodology based on this algorithm for head motion detection and subsequent event-based correction using external markers. Use of markers has previously been studied for similar applications [29, 30].

## 2 Materials and methods

### 2.1 Overview

The proposed method has two steps, namely, motion detection and correction. For motion detection, listmode data is sorted into short, fixed duration time frames of 500ms. The TOF-PEPT algorithm is then applied to determine when global motion occurred during the data acquisition. Subsection 2.2 shows the use of TOF-PEPT to extract global head movement information. Based on the global head position, it is possible to define transitional segments where the patient was in motion, and static segments represent where the patient remained near motionless as shown in Fig 1. Subsection 2.3 describes how the motion free static frames are identified. Subsequent motion correction can be achieved either by means of reconstructed image registration or transformation of the LORs. This work focuses on transformation of the LORs using weak radioactive point sources placed on radiolucent glasses worn by the patient. Subsection 2.4 describes a counting based TOF-weighted line density algorithm for locating the point sources based on which transformations between two static frames can be obtained. Subsection 2.5 describes the correction of individual LOR.

### 2.2 Motion signal extraction: TOF-PEPT

TOF-PEPT [28] is a modified version of the PEPT algorithm originally proposed by Parker et al. [31] The algorithm determines the location of the centroid of an LOR distribution within a region-of-interest (ROI) by iteratively estimating the point in space $p_m$ that minimizes a

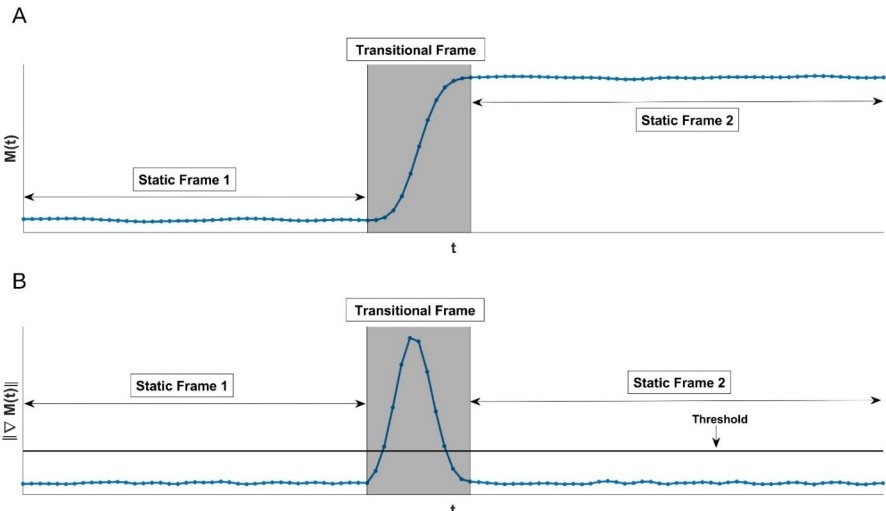

**Fig 1. Motion frame detection.** Illustration of detecting static frames: (A) Estimated motion (B) Gradient approximation of motion. The sequence of short duration time frames was divided into transitional frames and longer duration static frames based on the data-driven threshold.

weighted sum of LOR related distances. Mathematically,

$$p_m = argmin_p \sum_{L_i \in \Omega} w_i \delta^2(L_i, p)$$

$$w_i \triangleq 1 + max\left\{0, 1 - \left(\frac{||p_i - p_m||}{\sqrt{2}\sigma_T}\right)^2\right\}$$

Here, $w_i$ is a weighting factor for each LOR, $\delta(L_i, p)$ represents the perpendicular distance from the LOR denoted by $L_i$ to some point p, $\Omega$ is the set of LORs considered, $p_i$ denotes the TOF estimate of the annihilation location for $L_i$, and $\sigma_T$, which represents the uncertainty associated with the TOF estimate, depends on the coincidence timing resolution of the scanner.

At the end of each iteration, LORs for which $\delta(L_i, p_m)$ were greater than the mean thereof plus one standard deviation were discarded. The remaining set of LORs were used to update the estimate of $p_m$ during the next iteration. The iteration stopped when the number of LORs left fell below the number of LORs whose annihilation locations were within a sphere of radius $k\sigma_T$ surrounding $p_m$. Parameter k controls the number of LORs to be kept, with higher k resulting in a larger number of retained LORs. For the results reported below, we used k = 2.

## 2.3 Static frame detection

With the application of TOF-PEPT, it was possible to measure the left-right, anterior-posterior, and superior-inferior movement. For ease of notation, we will henceforth refer to this movement as changes in X, Y and Z coordinates. Let M(t) represent the estimated motion. We then used a forward difference approximation of the gradient, i.e., $\nabla M(t) = M(t+1) - M(t)$, and its norm thereof was calculated as follows:

$$||\delta M(t)||_2 = \sqrt{|X(t+1) - X(t)|^2 + |Y(t+1) - Y(t)|^2 + |Z(t+1) - Z(t)|^2}$$

The resulting signal was smoothed, and a threshold was used to determine when motion had taken place. The threshold was set to be a scalar λ times the mean absolute deviation (MAD) of the smoothed gradient norm calculated for the full duration of the scan. Mathematically, MAD can be expressed as follows:

$$\text{MAD} = \frac{1}{n}\sum_{t=1}^{n}|S_t - \bar{S}| \quad \text{where} \quad S_t = ||\delta M(t)||_2$$

Here, $n$, $S$, and $\bar{S}$ represent the number of sampling data points, the data values of the signal of interest, and their average, respectively. Consecutive sequences of short duration time frames for which the threshold was exceeded were considered transitional and ultimately discarded (as the patient's head was in motion) while those for which the threshold was not exceeded were combined into longer duration so-called static time frames. See Fig 1 for an illustration. For the results reported below, we used λ = 3 which we found to allow all significant patient movement to be detected. The mean absolute deviation between two consecutive time points was $0.\tilde{5}$ mm for the experimental data studied resulting in a threshold of $1.\tilde{5}$ mm.

## 2.4 Point source detection: TOF-weighted line density method

Three or more point sources were used to guarantee unique transformations be obtained. Several techniques have been proposed for locating multiple particles in listmode data [32–34]. Here we propose an approach centered on a TOF-weighted line density method.

An accumulation array was created that covered the field-of-view with a spatial resolution corresponding to that of the scanner. In principle, all voxel values could be initialized to 0 and then incremented based on the LOR intersections. In practice, each LOR was backprojected using Bresenham's algorithm [35] using a TOF centered Gaussian distribution with a FWHM matching the timing resolution of the scanner. The resultant grid count approximated the distribution of the LORs within the field-of-view.

The point sources were distinguishable in the accumulation array as regions with more dense LOR distributions compared with the brain uptake regions. Assuming that N point sources produce N local maxima, an iterative search was carried out locating them one at a time. First, the voxel with the highest count was identified. That provided a location estimation of the corresponding point source. To further refine the estimation, a weighted average of the voxel coordinates within a neighborhood of k × k × k was taken with the voxel values as weights. If $x_{max}, y_{max}, z_{max}$ denotes the coordinates of the voxel with highest count, and Ω denotes the neighborhood region centered on it, then the estimation of true location was made by:

$$x_{est} = \frac{\sum_{i\in\omega} v_i x_i}{\sum_{i\in\omega} v_i}$$

$$y_{est} = \frac{\sum_{i\in\omega} v_i y_i}{\sum_{i\in\omega} v_i}$$

$$z_{est} = \frac{\sum_{i\in\omega} v_i z_i}{\sum_{i\in\omega} v_i}$$

Here, $(x_i, y_i, z_i)$ denotes the voxel coordinates in the neighborhood and $v_i$ denotes their corresponding values. The neighborhood counts were reset to 0 before continuing the search for the next point source. The value of k was chosen to be large enough to cover the point sources

uptake regions, yet small enough to not exceed the minimum distance between two point sources. Empirically, we found k = 9 to produce good results for our data.

## 2.5 Rigid transformation: Calculation and application

The rigid transformations needed to register the static frames to the reference frame were obtained using SVD-based Procrustes analysis [36]. For a rigid body motion, transformation from position (x, y, z) to position (x', y', z') can be realized by a rotation followed by translation as follows:

$$
\begin{bmatrix} x' \\ y' \\ z' \end{bmatrix} = \begin{bmatrix} R_{xx} & R_{xy} & R_{xz} \\ R_{yx} & R_{yy} & R_{yz} \\ R_{zx} & R_{zy} & R_{zz} \end{bmatrix} \begin{bmatrix} x \\ y \\ z \end{bmatrix} + \begin{bmatrix} t_x \\ t_y \\ t_z \end{bmatrix}
$$

where, Rxx, Rxy, Rxz, Ryx, Ryy, Ryz, Rzx, Rzy, Rzz are rotational parameters, and tx, ty, tz are translational parameters. The spatial coordinates of an LOR can be re-positioned to a reference position by applying the above transformation. In this work, the first static frame of the PET data served as the reference to which subsequent frames were registered, however, the algorithm allows for any user-defined frame of reference to be chosen. The static frame LORs were repositioned and the TOF information updated accordingly. Repositioned LORs that did not match up with physical detector crystals were discarded. The remaining LORs were merged into a new listmode data set from which a final motion-free image was reconstructed.

## 2.6 Data and validation

Five patients were recruited and and written consent obtained for all participants at our outpatient PET/CT facility under the auspices of the University of Tennessee, Knoxville Institutional Review Board approved protocol (IRB #3941). Imaging was performed on a 64-slice Biograph mCT Flow PET/CT using full 64-bit listmode data acquisition. The patients were asked to rest their head in random positions and orientations at different time points during a 3-minute scan. Each study thus exhibited a different range of movements and therefore produced a different number of static frames. The whole field-of-view (FOV) was selected as ROI for motion detection by TOF-PEPT. The full patient imaging workflow proceeded as follows: 1) standard of care PET/CT imaging was performed and finalized; 2) point sources were positioned onto the patient;3) listmode acquisition of the head motion sequence was performed. As mentioned above, point sources were used to obtain the motion correction transformation parameters. Three or four point sources were placed on a pair of radiolucent plastic lab goggles worn by the patient with each source positioned to be off axis from one another. Typically, one source was placed on each side of the goggles, and one placed on the front near the forehead of the patient. Fig 2 shows the apparatus used. We used 74-185 kBq (2-5 μCi) point sources in the form of zeolite beads 2 mm in diameter. Motion corrected listmode data were rebinned into sinograms from which attenuation and scatter corrected images were reconstructed using the OSEM+TOF algorithm available on the Biograph mCT Flow PET/CT scanner. For reconstruction we used our institution standard clinical protocol with 3 iterations, 24 subsets and 5 × 5 Gaussian post-smoothing. All data processing and reconstruction was performed using the Siemens e7 processing tools or the built-in scanner histogramming and reconstruction algorithms (Siemens Healthineers, Knoxville).

To comparatively evaluate the performance of our automated LOR motion correction, we implemented a more conventional image-based method centered on manual landmark-based

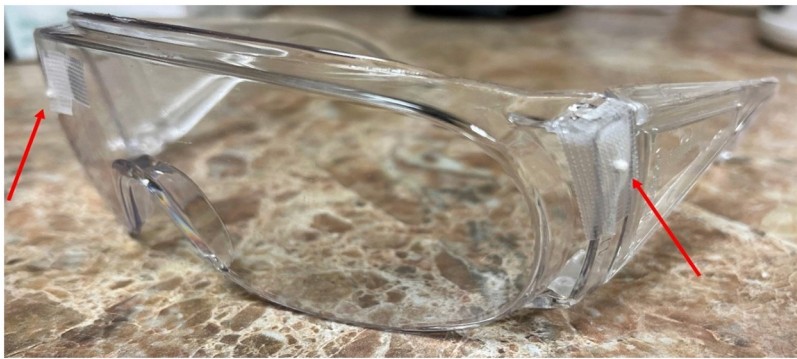

**Fig 2. Goggles apparatus.** Radiolucent plastic lab goggles worn by the patient with point sources attached to it. Two sources were placed on two sides of the goggles, and one placed on the front near the forehead of the patient. In the figure, two point sources can be seen to be indicated by two red arrows.

registration of volumetric images reconstructed for each static frame detected by our motion tracking algorithm. More specifically, manually drawn regions-of-interest covering each source were subjected to thresholding (15% of maximum value) to create a voxelized mask where the centroid of the ROI could be found and used as the series of landmarks for each frame. The attenuation corrected frame images were aligned and summed to produce a single motion corrected image.

In addition to our quantitative comparisons, a qualitative review was performed by two board certified radiologists, both with more than ten years of clinical PET diagnostic assessment experience. Each reviewed the case studies and visually compared the manually landmark corrected images to the automated algorithm proposed in this work. Their visual assessments were used to determine if any significant artifacts had been introduced into the automatically processed images and whether the resulting corrected images were diagnostically useful.

## 3 Results

### 3.1 Estimated motion and static frames

Fig 3 shows the head displacement as a function of time for the five patient studies. They are shown along the principal direction of motion, i.e., the direction with the largest range of movement. The dominant movement was observed in the XY (axial) and ZX (coronal) planes for all the studies since the patient head movement was mostly around Z and Y axes. Patient 1 moved twice, and the head was thus at rest at three different positions with three static frames produced correspondingly. Patient 2 moved three times leading to four static frames. Patients 3, 4 and 5 produced three, four, and four static frames, respectively. The shaded grey regions in Fig 3 represent transitional frames. The head movements ranged from less than 5 mm to as much as 50 mm.

### 3.2 Evaluation of transformation parameters

Fig 4A and 4C plot the axial and coronal plane locations of the three point sources used for one patient study. Fig 4B and 4D show the effect of having applied the transformation parameters. Visually, the point sources in frames 2 and 3 are seen to have been registered well to the point sources in frame 1 which served as the reference. We calculated the mean Euclidean

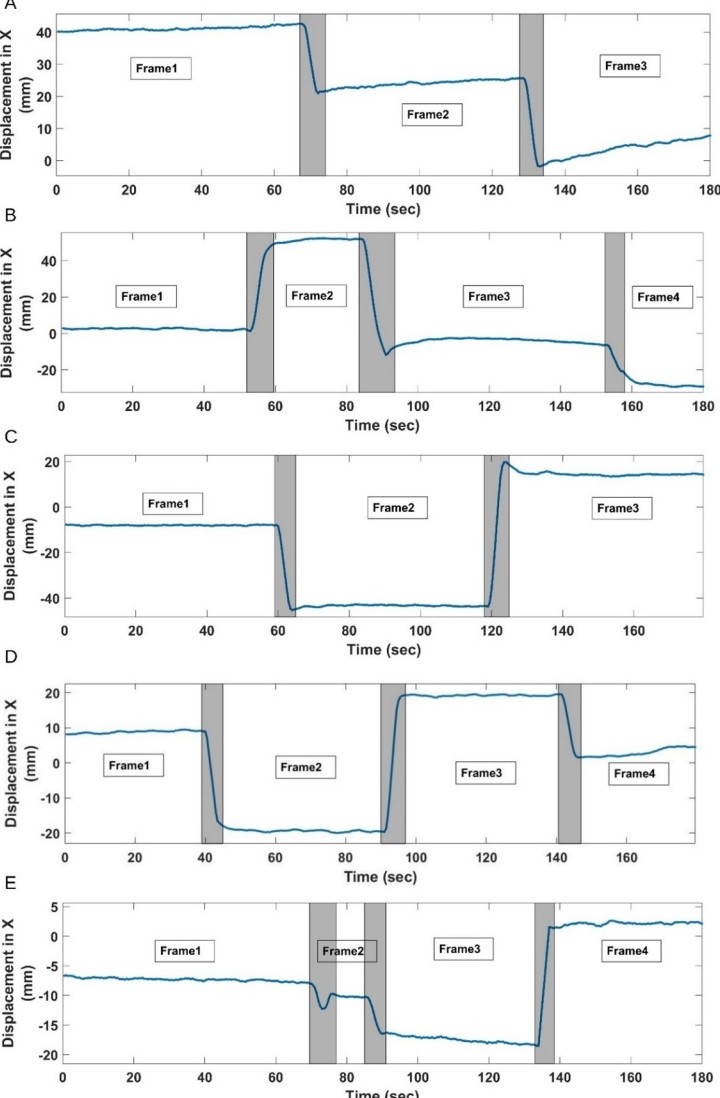

**Fig 3. Motion segment extraction.** Plots of motion in principal direction along with the estimated transitional and static frames: (A) Patient Study 1: Three static frames; (B) Patient Study 2: Four static frames; (C) Patient Study 3: Three static frames; (D) Patient Study 4: Four static frames; and (E) Patient Study 5: Four static frames.

distance between the reference and transformed location coordinates for all five patient studies. Table 1 shows the results which indicate that the registration was within a mean distance of 1.2±0.09 mm.

## 3.3 Comparison of motion corrected images

Fig 5 shows the comparison of the motion corrected images in the axial plan for Patient Study 3 (Top row) and Patient Study 5 (bottom row). The left column shows a slice of the original scan with full patient motion showing the impact on image quality. The head can be seen to be positioned in three different orientations resulting in significant blurring and obscuring of detail. The initial head position was chosen as the first static reference frame. The middle

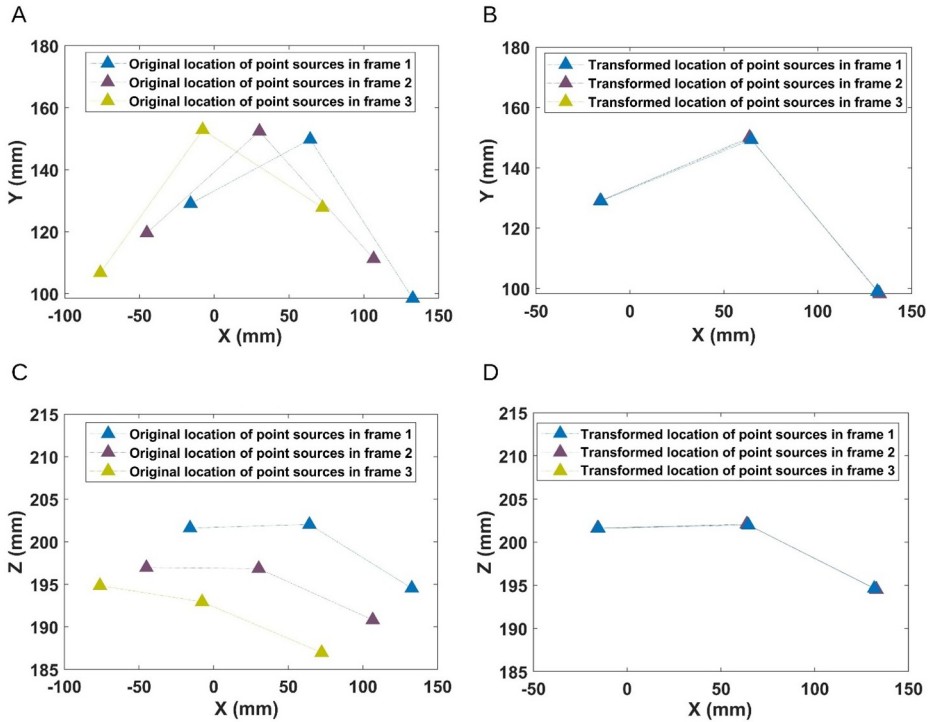

**Fig 4. Comparison of original point source positions with transformed locations.** Point source locations in each different frame in the axial plane (top row) (A) before and (B) after application of the transformation, and in the coronal plane (bottom row) (C) before and (D) after application of the transformation.

column shows a slice from the manually corrected image volume using the comparison standard based on image-based landmark registration. The right column shows our automated data-driven correction result where the static frames of motion were detected, transformation matrices were calculated and applied to the individual LORs, and then histogrammed and reconstructed. Fig 6 shows the sagittal and coronal view respectively for Patient Study 3 and Patient Study 5. The motion corrected images show that the shifted head positions were correctly repositioned to the reference position.

To quantitatively measure the alignment, we used the Jaccard similarity index given by

$$J(A, B) = \frac{|A \cap B|}{|A \cup B|}$$

**Table 1. Mean Euclidean distance between reference and transformed point source coordinates.**

| Study | Mean Euclidean Distance (mm) | | |
|---|---|---|---|
| | **Frame 2** | **Frame 3** | **Frame 4** |
| Patient Study 1 | 1.48 | 1.36 | — |
| Patient Study 2 | 2.31 | 2.51 | 0.78 |
| Patient Study 3 | 0.52 | 0.77 | 0.73 |
| Patient Study 4 | 0.38 | 0.77 | — |
| Patient Study 5 | 1.24 | 0.78 | 1.68 |
| Mean | 1.17 | 1.23 | 1.06 |

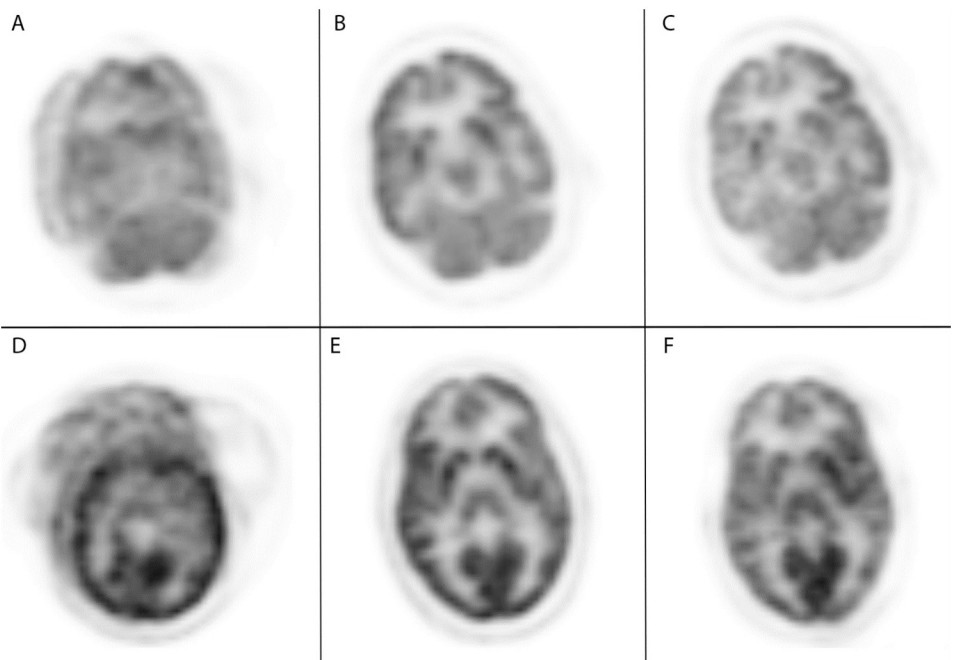

**Fig 5. Reconstructed axial comparisons of uncorrected data, frame-based corrected data, and event-based corrected data.** Reconstruction example for Patient Study 3 (Top row) and Patient Study 5 (Bottom row) in the axial plane: (A, D) original motion impacted data; (B, E) frame-based correction; (C, F) event-based correction. Both datasets were set to a window of 4% of the maximum value.

where A and B respectively denote a reference image and a registered image. By design, $0 \leq J(A,B) \leq 1$ with a large value indicating a high degree of similarity. We drew a region-of-interest covering the brain, then subjected it to 15% and 25% thresholding to create two sets of masks to compare. Table 2 lists the resulting Jaccard similarity indices for all five patient studies. It can be seen that the frame-by-frame image registration-based gold standard created for this study and our automated LOR based correction achieved similar results with number in the range of 92.5±4.8% for the former and 93.2±4.5% for the latter.

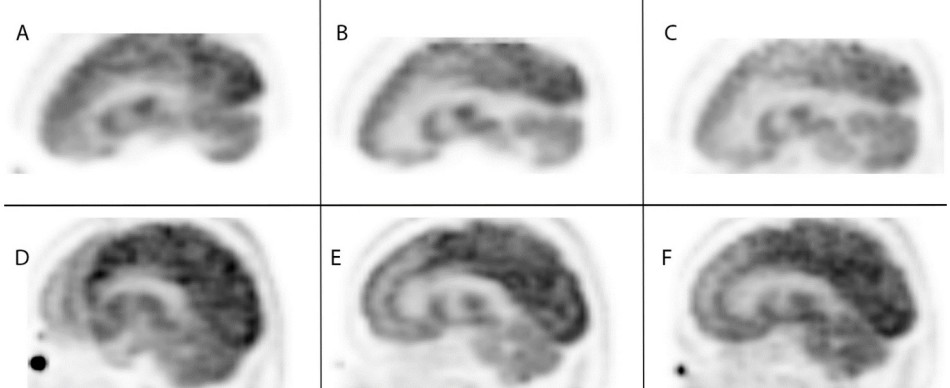

**Fig 6. Reconstructed saggital comparisons of uncorrected data, frame-based corrected data, and event-based corrected data.** Reconstruction example for Patient Study 3 (Top row) in sagittal plane and Patient Study 5 (Bottom row) in the coronal plane: (A, D) original motion impacted data; (B, E) frame-based correction; (C, F) event-based correction.

**Table 2. Quantitative evaluation of corrected image.**

| Study | Threshold | Jacard Index (%) | |
|---|---|---|---|
| | | Event Based | Frame Based |
| Patient Study 1 | 15% | 94.2 | 95.0 |
| | 25% | 85.2 | 88.4 |
| Patient Study 2 | 15% | 97.9 | 96.9 |
| | 25% | 91.3 | 90.3 |
| Patient Study 3 | 15% | 98.6 | 98.9 |
| | 25% | 96.1 | 97.1 |
| Patient Study 4 | 15% | 94.4 | 93.7 |
| | 25% | 87.1 | 84.8 |
| Patient Study 5 | 15% | 94.7 | 96.1 |
| | 25% | 86.0 | 91.2 |
| Mean | | 92.5 | 93.2 |

Qualitative diagnostic review of the images indicated that the manufactured "gold-standard" manually registered images had slightly better, but comparable, image quality to that of the automated algorithm. The difference that the radiologists noted between images from two methodologies was localized regional changes in the reconstructed activity concentration between the two methodologies. The research PET studies for this work were acquired after the patient's standard of care diagnostic PET/CT studies.

## 4 Discussion

We have described a data-driven methodology that can extract motion information from list-mode data and perform event-based correction without the need for reconstructing and aligning multiple image frames. Post data acquisition, the proposed approach can be applied to raw listmode data in a fully automated way without the need for manual intervention. The current implementation corrects for types of motion that occur during routine patient imaging, such as coughing, sneezing, falling asleep, etc. These are typically rather abrupt with short phases of motion followed by periods where the patient is in a static state. The static frame detection could be extended to take slow motion into account by analyzing the data at a different time scale.

The proposed method consists of two steps. First, motion detection is achieved using our previously published TOF-PEPT algorithm which is capable of detecting head movement directly from the LOR data independent of any markers. Second, motion correction is carried out in one of two ways. As shown in this paper, external point sources in the form of radioactive beads attached to glasses worn by the patient facilitate correction of the listmode data which in turn allows subsequent reconstruction of a single volume image from the transformed LORs. Alternatively, frame-based reconstructed image registration can be carried out using the static frames detected by the TOF-PEPT algorithm followed by summation of manually registered images. This provides good flexibility of the algorithm to operate within a number of different clinical imaging scenarios and offers the ability for retrospective analysis and correction so long as the point sources are on the patient during the scan.

The proximity of the point sources warranted some concern for possible exposure to the sensitive lens of the eye even though the point sources used in this work were very weak. As part of our initial IRB review and approval, we estimated lens doses to the eye under worst-case conditions using a 5 microCurie source positioned 1 cm directly in front of the eye

without accounting for beta shielding from the plastic goggles. Our total dose rate estimated under these worst case conditions was 14.3 mGy/hr which even for 30 minutes of total exposure would be 70x less than even the updated 0.5 Gy deterministic limit set by the International Commission on Radiation Protection in 2018. Our more realistic, yet still conservative, estimates assuming a 5 microCurie source, goggle beta shielding, 1 cm distance, and 15 minutes of exposure resulted in a total estimated dose of 0.07 mGy to the lens of the eye. Our TOF--PEPT based motion detection algorithm was configured to use a sufficiently high time resolution (500ms frames) to minimize intra-frame motion. We showed that the algorithm detected all notable patient movements for all patient studies. For our proposed event-based motion correction, external point sources were used as unique data points to find three-dimensional rotation and translation information. We achieved reliable transformation parameters with sufficient accuracy such as mean Euclidean distance between reference and transformed location that allowed repositioning of shifted LORs to the reference position.

All data were corrected for attenuation and scatter using standard manufacturer correction methods. This was possible because the corrected LORs were rotated to the CT orientation enabling use of built-in processes. A comparison of scatter, randoms, and normalization corrections on the rotated LOR positioning versus original orientation was outside the scope of this work, but none of the reconstructed images showed significant signs of degradation due to such artifacts. We hypothesize that limited artifacts were observed in our results due to fairly uniform and symmetric attenuation, scatter, and uptake in the brain and head; analogous to the assumptions when performing Chang's attenuation correction for SPECT imaging [37]. Future work is planned to assess the impact of rotating LORs with and without rotating the associated correction factors for the specific head imaging case.

To estimate location of the multiple point sources, we proposed a variation of the PEPT technique, following a line density-based algorithm and utilizing TOF information. In our previous implementation of TOF-PEPT for respiratory motion correction, we tracked multiple particles by drawing individual ROI for each particle and limiting the tracking only to the corresponding ROI. However, for large motion such as head/whole body motion, having an ROI that will capture the entire range of movement is not straightforward unless each particle is labelled at different levels of radioactivity uptake. Hence for the purpose of motion correction, we adopted this variation of back-projection. It will be our future endeavor to extend the application of TOF-PEPT method to track multiple particles alleviating the constraints on radioactivity uptake.

As part of this work, we studied the impact of the kernel size for tracking multiple sources. It was observed that the smaller the kernel size, the higher the error while using it for resetting the neighboring voxel values. It was speculated that with smaller kernel size, all the voxels corresponding to one particular point source were not properly taken into account and thus reset to continue the search for next point source. Thus, a desirable kernel size should be large enough to cover the region of the point source and smaller than half the minimum distance between two point sources. To note, the size of kernel in calculation of weighted average of voxel coordinates did not result in much difference and we used same kernel for both purposes. When dealing with external point sources, they are expected to have sufficient activity concentration compared to the brain area so that higher valued voxels in the accumulation array correspond to the point sources. This was easily achieved with reasonably low radioactive point sources of small sizes which had higher activity concentrations and did not interfere with diagnostic image quality. We tested different surfaces on which the point sources can be placed including a head band and directly on the skin but found lab goggles to be more comfortable for the patient while also providing a secure fit. Using a structure to separate the source from the patient skin is desirable as it enables an air gap between the activity naturally

being taken up by the patient tissue and the sources being tracked. For this study, the patients were told when to move so we knew when to visually observe for any possible glasses movement. Additionally, the ranges of motion assessed also did not result in head positioning that caused contact with the goggles that would have caused unwanted movement and would have resulted in an additional PET acquisition. Only visual assessment of motion was performed but we did not observe movement of the glasses relative to the head for any of the patients, however, one limitation of using external sources that are not attached to the skin would be the potential risk for the glasses to shift during imaging in such way as to no longer be aligned with the patient head movement.

To qualitatively and quantitatively evaluate the performance of our proposed approach in motion correction, we created motion corrected images both from our automatically corrected listmode data and by applying the frame-by-frame reconstructed image registration technique. The frame-based landmark registration technique was used as a gold standard with which to test our automated motion detection and correction accuracy. Our Jaccard similarity index results show our automated technique to be nearly identical to the manual landmark registration method. Furthermore, this manual landmark registration technique can only be used if precise knowledge of when the motion occurred during acquisition is known. That precise information was derived from our automated motion detection methodology and knowledge of when the motion occurred during imaging would typically not be available in routine clinical practice without some other form of patient monitoring.

Additional qualitative review of the radiologists found the performance of our proposed approach comparable to the manually registered approach. Both methods showing similar performance, the results favor the automated approach as it eliminates the need to reconstruct multiple image volumes and perform manual data correction with the added benefit of being able to automatically determine whether significant motion occurred during imaging and then correct the data accordingly.

## 5 Conclusion

In previous work, we studied respiratory motion correction and proposed a TOF weighted PEPT algorithm. In this paper, we applied the algorithm to detect head motion and showed that the algorithm can detect such motion reliably from the raw listmode data. We performed event-based correction of the raw data by deriving transformation parameters with the application of external markers and a line density method. The motion corrected data showed significant improvement over the originally motion impacted data, validated by the images reconstructed from the corrected data. Our motion corrected data also was able to create nearly equivalent qualitative results compared to labor intensive, but accurate, manual landmark registration techniques.

## Author Contributions

**Conceptualization:** Dustin R. Osborne.

**Data curation:** Dustin R. Osborne.

**Formal analysis:** Tasmia Rahman Tumpa, Jens Gregor, Yong Bradley, Yitong Fu, Dustin R. Osborne.

**Investigation:** Tasmia Rahman Tumpa, Jens Gregor, Dustin R. Osborne.

**Methodology:** Tasmia Rahman Tumpa, Jens Gregor, Dustin R. Osborne.

**Software:** Tasmia Rahman Tumpa.

**Supervision:** Jens Gregor, Dustin R. Osborne.

**Writing – original draft:** Tasmia Rahman Tumpa, Jens Gregor, Dustin R. Osborne.

**Writing – review & editing:** Tasmia Rahman Tumpa, Jens Gregor, Yong Bradley, Yitong Fu, Dustin R. Osborne.

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
