## [Decision Letter · Decision Letter 0]

15 Jun 2022

PONE-D-22-14720Data-Driven Head Motion Correction for PET using Time-of-Flight and Positron Emission Particle Tracking TechniquesPLOS ONE

Dear Dr. Osborne,

Thank you for submitting your manuscript to PLOS ONE. After careful consideration, we feel that it has merit but does not fully meet PLOS ONE’s publication criteria as it currently stands. Therefore, we invite you to submit a revised version of the manuscript that addresses the points raised during the review process.

We look forward to receiving your revised manuscript.

Kind regards,

Jonathan Engle, PhD

Academic Editor

PLOS ONE

Journal Requirements:

2. Please ensure you have stated in the Ethics Statement on the online submission form whether participant consent was informed (via 'Edit Submission'). Please also ensure you state in the Methods section of your manuscript text this information regarding participant consent.

Additionally, please note that PLOS ONE has specific guidelines on code sharing for submissions in which author-generated code underpins the findings in the manuscript. In these cases, all author-generated code must be made available without restrictions upon publication of the work. Please review our guidelines at https://journals.plos.org/plosone/s/materials-and-software-sharing#loc-sharing-code and ensure that your code is shared in a way that follows best practice and facilitates reproducibility and reuse.

Additional Editor Comments:

The manuscript is a sound description of a method for motion correction in PET, and the reviewer points out the two main concerns I had after my original read of the submission. Please address these in your response.

Reviewers' comments:

Reviewer's Responses to Questions

**Comments to the Author**

1. Is the manuscript technically sound, and do the data support the conclusions?

Reviewer #1: Yes

2. Has the statistical analysis been performed appropriately and rigorously? 

Reviewer #1: N/A

3. Have the authors made all data underlying the findings in their manuscript fully available?

Reviewer #1: Yes

4. Is the manuscript presented in an intelligible fashion and written in standard English?

Reviewer #1: Yes

5. Review Comments to the Author

Reviewer #1: There are two concerns which I think need to be addressed. The first is the dosimetry of the point sources to the lens of the eye. This is one of the critical organs and the dose here is not addressed even though the point sources on the glasses sit very close to the eye. The activity level in the point sources was mentioned in order to be able to do the estimate. The other concern is in the movement of the glasses with respect to the skull. It was mentioned that there was no movement, but it was not mentioned if this was solely by visual inspection or if there was another method used.

6. PLOS authors have the option to publish the peer review history of their article (what does this mean?). If published, this will include your full peer review and any attached files.

Reviewer #1: No

---

## [Author Response · Author response to Decision Letter 0]

6 Jul 2022

Point by Point Response to Reviewers

1. Reviewer feedback: Point source dosimetry and discussion not included in manuscript

Point source dosimetry information was added to the discussion section and more detail and context also provided in our cover letter.

Cover Letter Text:

Regarding point source dosimetry, this was a concern when we initiated the Institutional Review Board process and we assessed lens dose to the eye in the worst-case scenario with our point sources fully exposed to the eye at 1 cm assuming also no goggles to provide beta shielding. The rate constants at 1 cm for F-18 is approximately 6 R/hr/mCi and 300 R/hr/mCi for gamma and beta exposure, giving a total exposure rate for our scenario of 306 R/hr/mCi. The maximum point source activity used was 5 microCuries giving a total exposure rate of 1.53 R/hr. Using the standard diagnostic conversion of 9.33 mSv/R we have a dose rate of 14.3 mSv/hr, which for gammas and betas equates to a maximum dose rate of 14.3 mGy/hr. 

Even in our extreme scenario, we determined we were still well under the very conservative 0.5 Gy acute eye lens deterministic limits set by the ICRP in 2018 that even with 30 minutes of exposure to the point sources would still be approximately 70x less than a dose that would deterministically cause any issues with the lens of the eye. More realistically, the ~5 mm of plastic likely shields nearly all of the betas and the point source distances were 2-4 cm from the lens of the eye having to also transmit through other materials (skull, etc.) to reach the lens. We determined using the assumptions below that our more realistic worst-case dose to the lens of the eye for a single point source used in this work was 0.07 mGy which is 7,000x less than the conservative ICRP guidance for deterministic lens effects:

• Beta emissions almost completely absorbed by 5 mm plastic goggles.

• Point source 1 cm from lens of the eye positioned directly in front of the eye.

• Total exposure of 15 minutes to the point sources

Updates to Discussion Section of Manuscript:

The proximity of the point sources warranted some concern for possible exposure to the sensitive lens of the eye even though the point sources used in this work were very weak. As part of our initial IRB review and approval, we estimated lens doses to the eye under worst-case conditions using a 5 microCurie source positioned 1 cm directly in front of the eye without accounting for beta shielding from the plastic goggles. Our total dose rate estimated under these worst case conditions was 14.3 mGy/hr which even for 30 minutes of total exposure would be 70x less than even the updated 0.5 Gy deterministic limit set by the International Commission on Radiation Protection in 2018. Our more realistic, yet still conservative, estimates assuming a 5 microCurie source, goggle beta shielding, 1 cm distance, and 15 minutes of exposure resulted in a total estimated dose of 0.07 mGy to the lens of the eye. 

2. Reviewer Feedback: Independent Goggle Movement from Patient head

Additional information on the goggles and our testing was added to our goggles/glasses discussion in the discussion section.

We tested different surfaces on which the point sources can be placed including a head band and directly on the skin but found lab goggles to be more comfortable for the patient while also providing a secure fit. Using a structure to separate the source from the patient skin is desirable as it enables an air gap between the activity naturally being taken up by the patient tissue and the sources being tracked. For this study, the patients were told when to move so we knew when to visually observe for any possible glasses movement. Additionally, the ranges of motion assessed also did not result in head positioning that caused contact with the goggles that would have caused unwanted movement and would have resulted in an additional PET acquisition. Only visual assessment of motion was performed but we did not observe movement of the glasses relative to the head for any of the patients, however, one limitation of using external sources that are not attached to the skin would be the potential risk for the glasses to shift during imaging in such way as to no longer be aligned with the patient head movement.

---

## [Editor Report · Decision Letter 1]

26 Jul 2022

Data-Driven Head Motion Correction for PET using Time-of-Flight and Positron Emission Particle Tracking Techniques

PONE-D-22-14720R1

Dear Dr. Osborne,

We’re pleased to inform you that your manuscript has been judged scientifically suitable for publication and will be formally accepted for publication once it meets all outstanding technical requirements.

Kind regards,

Jonathan Engle, PhD

Academic Editor

PLOS ONE
---

## [Editor Report · Acceptance letter]

22 Aug 2022

PONE-D-22-14720R1 

Data-Driven Head Motion Correction for PET using Time-of-Flight and Positron Emission Particle Tracking Techniques 

Dear Dr. Osborne:

I'm pleased to inform you that your manuscript has been deemed suitable for publication in PLOS ONE. Congratulations! Your manuscript is now with our production department. 

Kind regards, 

on behalf of

Dr. Jonathan Engle 

Academic Editor

PLOS ONE